# Population pharmacokinetic modelling of primaquine exposures in lactating women and breastfed infants

Thanaporn Wattanakul [1], Mary Ellen Gilder[2], Rose McGready [2,3], Warunee Hanpithakpong[1], Nicholas P. J. Day[1,3], Nicholas J. White [1,3], François Nosten [2,3], Joel Tarning [1,3] & Richard M. Hoglund [1,3] ✉

Current guidelines advise against primaquine treatment for breastfeeding mothers to avoid the potential for haemolysis in infants with G6PD deficiency. To predict the haemolytic risk, the amount of drug received from the breast milk and the resulting infant drug exposure need to be characterised. Here, we develop a pharmacokinetic model to describe the drug concentrations in breastfeeding women using venous, capillary, and breast milk data. A mother-to-infant model is developed to mimic the infant feeding pattern and used to predict their drug exposures. Primaquine and carboxyprimaquine exposures in infants are <1% of the exposure in mothers. Therefore, even in infants with the most severe G6PD deficiency variants, it is highly unlikely that standard doses of primaquine (0.25-1 mg base/kg once daily given to the mother for 1-14 days) would cause significant haemolysis. After the neonatal period, primaquine should not be restricted for breastfeeding women (Clinical Trials Registration: NCT01780753).

Primaquine is recommended for the radical cure of *Plasmodium vivax* and *Plasmodium ovale* malaria as it kills the latent liver stage hypnozoites that cause relapses. Primaquine is also used to block the transmission of falciparum malaria as it kills gametocytes, the stage of the parasite life cycle that is transmitted to the mosquito.

Current guidelines advise against giving primaquine to breastfeeding mothers because of potential haemolytic toxicity if the breastfeeding infant has glucose-6-phosphate dehydrogenase (G6PD) deficiency[1]. In areas with frequent pregnancies and where infants are breastfed for long periods, this could result in mothers being denied adequate treatment for malaria. This is the situation along the malaria-endemic border between Thailand and Myanmar, where prolonged breastfeeding is common[2]. In this area, vivax malaria relapses in approximately 50% of those who do not receive a radical cure[3]. Current guidelines for primaquine in pregnant and breastfeeding mothers mean that approximately 13% of females are excluded from radical cure because of these restrictions. If the

restriction against breastfeeding was removed, this could be reduced to 4%[4].

Primaquine is an antimalarial drug with a relatively short half-life (5-6 h) and complex metabolism that is still incompletely characterised[5]. The major metabolite carboxyprimaquine, thought to be inactive, is formed by metabolism via monoamine oxidase[6,7]. The active unknown metabolite or metabolites are generated by the cytochrome P450 enzymatic pathway, in which CYP2D6 plays a prominent role[7]. The estimated oral bioavailability of primaquine is high (96%) and the plasma protein binding is reported to be >90% with alpha 1-acid glycoprotein as the main binding protein[5,8]. The population pharmacokinetics of primaquine have been well described in both children and adults[9–13].

Primaquine ingestion results in haemolysis in G6PD-deficient individuals. The severity of primaquine-induced haemolysis depends on the daily dose administered, the duration of administration, and the severity of the G6PD deficiency[14,15]. The mechanism of the haemolytic

[1]Mahidol Oxford Tropical Medicine Research Unit, Faculty of Tropical Medicine, Mahidol University, Bangkok, Thailand. [2]Shoklo Malaria Research Unit, Mahidol Oxford Tropical Medicine Research Unit, Mahidol University, Mae Sot, Thailand. [3]Centre for Tropical Medicine and Global Health, Nuffield Department of Medicine, Oxford University, Oxford, UK. ✉e-mail: richard.hoglund@ndm.ox.ac.uk

effect of primaquine has been studied both in vitro and in vivo[16–18]. The CYP2D6-dependent phenolic metabolite (5-hydroxyprimaquine) is likely to be involved in the mechanism of primaquine-induced haemolysis through highly reactive transient metabolite species or through an oxidative by-product (e.g. hydrogen peroxide) from redox cycling[18–20]. The enzyme activity and, thus, genetic polymorphisms of CYP2D6 may affect the degree of haemolysis. For infants, the maturation of the CYP2D6 enzyme remains incomplete until they reach the age of one year[21,22]. Therefore, the biotransformation of primaquine into haemolytic metabolites is expected to be lower in infants compared to adults or older children. Carboxyprimaquine is not thought to be biologically active and is, therefore, not implicated in the mechanism of haemolysis.

G6PD testing for neonates is often unavailable in malaria-affected communities, and the result may be difficult to interpret because of the haematological dynamics at birth[23,24]. The unknown G6PD status and, thus, potential vulnerability of the breastfed infant is, therefore, the main reason for not giving primaquine to breastfeeding women. To assess the haemolytic risk in the infant, it is important to know the amount of drug that infant receives from the mother. Only one study has investigated the pharmacokinetics of primaquine in plasma and breast milk. It showed that only a very small daily dose was received by the breastfeeding infant[25]. These data can be used to simulate different dosing scenarios and predict the total drug exposure in infants. The current study extends the previous analysis to develop a mechanistic pharmacokinetic model in order to predict the exposure of infants to primaquine and to evaluate the current treatment restrictions in lactating women.

## Results

### Pharmacokinetics in breastfeeding women

Twenty-one breastfeeding women were included in the pharmacokinetic analysis (Table 1). At first, the population pharmacokinetic model was developed using venous and capillary concentrations of primaquine and carboxyprimaquine. A one-compartment disposition model for both primaquine and carboxyprimaquine described the data satisfactorily. A two-compartment disposition model for primaquine did not improve the model fit (difference in objective function value; $\Delta OFV = -4.43$, degree of freedom; df = 4). For carboxyprimaquine, a two-compartment disposition model improved the model fit significantly ($\Delta OFV = -227$, df = 4). However, the model estimated a very low volume of distribution of central compartment of carboxyprimaquine ($V/F_{CPQ} \sim 4$ L) with a large inter-individual variability (IIV, coefficient of variation (%CV) > 100), which was considered to be physiologically impossible. The one-compartment disposition model of carboxyprimaquine did not show any systematic bias, thus it was carried forward for further model development.

A first-pass metabolism ($F_M$) was implemented in the model which significantly improved model fit ($\Delta OFV = -116$, df = 2). A transit absorption model with first-order absorption rate constant ($k_a$) set equal to transit absorption rate constant ($k_{tr}$, 4 transit compartments) improved the model fit compared to the traditional first-order absorption model ($\Delta OFV = -640$, df = 0). Estimating both $k_a$ and $k_{tr}$ resulted in an improved model fit ($\Delta OFV = -129$, df = 1), but the precision of the estimated $k_a$ was poor (relative standard error (%RSE) = 68.7) and the precision of all other pharmacokinetic parameters were decreased. With this model, the optimal number of transit compartments were also increased (12 transit compartments), resulting in a substantially increased run time of the model (~8 times longer). The goodness-of-fits and the prediction-corrected visual predictive checks of this model were comparable to the model where $k_a$ was set to be equal to $k_{tr}$. Thus, the simpler model was selected for further analysis.

Adding inter-occasion variability (IOV) on bioavailability (F) improved the model fit ($\Delta OFV = -211$, df = 1), and adding IOV on mean transit absorption time (MTT) improved the model fit

($\Delta OFV = -555$, df = 1). Adding IOV on both F and MTT further improved the model ($\Delta OFV = -844$, df = 2). Allometric scaling of body weight was implemented a priori on both clearance and volume of distribution ($\Delta OFV = 6.20$, df = 0). Although the allometric scaling of body weight did not improve the model fit, it was retained in the model based on previous physiological knowledge and to allow translational simulations using the developed model. The conversion factors between venous and capillary drug concentrations for both primaquine ($CF_{PQ}$) and carboxyprimaquine ($CF_{CPQ}$) were estimated to be 0.902 and 1.06, respectively. The IIV of these two conversion factors were estimated to be ≤ 10% and the precision of estimates was poor (%RSE, $IIV_{CF-PQ} = 45$ and $IIV_{CF-CPQ} = 139$). Fixing these IIV parameters to zero showed no systematic bias in the goodness-of-fits and had a minimal impact on all of other parameter estimates. Thus, this model was carried forward to evaluate the covariate effect using a stepwise covariate approach.

The effects of patient characteristics on pharmacokinetic parameters were evaluated using a stepwise covariate search approach. The significance level was defined at a $P$ value of <0.05 for forward inclusion and <0.001 for backward elimination. During the stepwise forward inclusion, smoking status was a significant covariate on relative bioavailability, where smokers had a 25.9% lower bioavailability compared to non-smokers ($\Delta OFV = -8.64$, df = 1). Additionally, age was a significant covariate on volume of distribution of primaquine ($V_{PQ}$) where the $V_{PQ}$ was increased by 1.29% per one-year increase of age ($\Delta OFV = -4.09$, df = 1). However, these two covariate effects were removed during the backward elimination step due to the more stringent criteria used.

The breast milk data were introduced to the final model. First, conversion factors between systemic mother venous concentrations and breast milk concentrations were estimated for both primaquine ($CFM_{PQ}$) and carboxyprimaquine ($CFM_{CPQ}$). The $CFM_{PQ}$ was estimated

**Table 1 | Demographics of breastfeeding women and infants**

| Demographics | Breastfeeding women (n = 21) | Infants (n = 21) |
|---|---|---|
| Age (years) | 23 (18–40) | 0.42 (0.13–1.81) |
| Sex (male/female) | 0/21 | 14/7 |
| Body weight (kg) | 51 (35–81) | 6.8 (4.13–10.8) |
| Haemoglobin level (mg/dL) | 12.3 (10.2–14.4) | 11.5 (10.5–14.2) |
| Haematocrit (%) | 37.7 (30.5–44.2) | 34.7 (33.0–43.2) |
| Total bilirubin level (mg/dL) | 0.365 (0.19–0.74) | - |
| Aspartate aminotransferase (U/L) | 25.5 (14–59) | - |
| Alanine aminotransferase (U/L) | 25.5 (7–54) | - |
| Alkaline phosphatase (U/L) | 95 (69–132) | - |
| Smoking status (yes) | 5 (23.8%) | - |
| Average number of breast-feeds (/day) | 11 (6–18) | - |
| Calculated infant daily milk intake (mL) | 1020 (619–1620) | - |
| Number of primaquine samples ( > LLOQ / < LLOQ) | | |
| Plasma | 542 / 7 | - |
| Capillary | 141 / 0 | 1 / 201 |
| Breast milk | 200 / 6 | - |
| Number of carboxyprimaquine samples ( > LLOQ / < LLOQ) | | |
| Plasma | 543 / 6 | - |
| Capillary | 143 / 0 | 73 / 129 |
| Breast milk | 133 / 73 | - |

Values reported as median (min-max) unless otherwise stated. *LLOQ* lower limit of quantification.

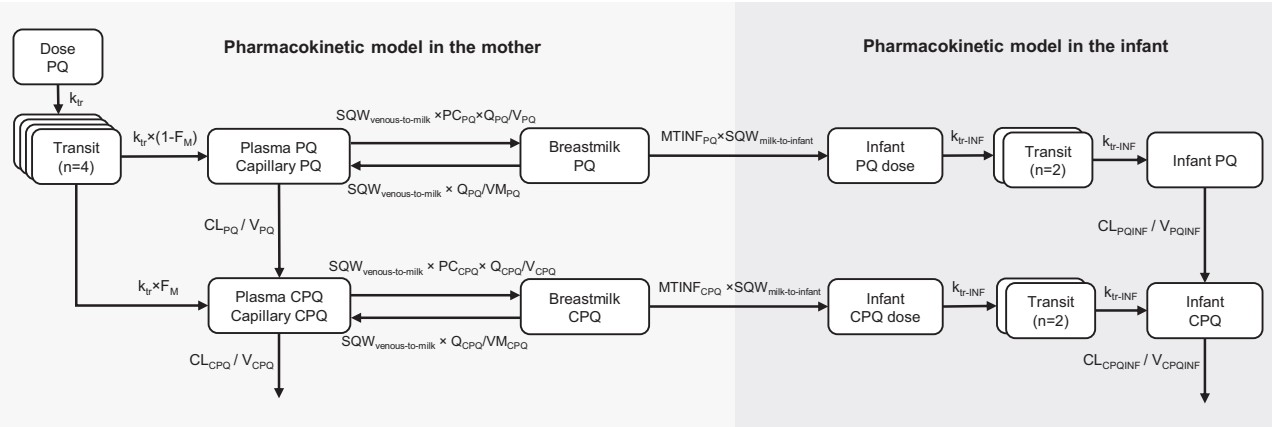

**Fig. 1 | A schematic figure of the pharmacokinetic model of primaquine-carboxyprimaquine.** The figure illustrating the structural pharmacokinetic model in the mother and the mother-to-infant model used to predict infant exposure. Abbreviations: PQ, primaquine; CPQ, carboxyprimaquine; n, number of transit compartments; $k_{tr}$, transit absorption rate constant in the mother; $F_M$, fraction of primaquine converted to carboxyprimaquine during first-pass metabolism; $CL_{PQ}$, apparent elimination clearance of primaquine; $V_{PQ}$, apparent central volume of distribution of primaquine; $CL_{CPQ}$, apparent elimination clearance of carboxyprimaquine; $V_{CPQ}$, apparent central volume of distribution of carboxyprimaquine; $PC_{PQ}$, fraction of primaquine that is freely distributed to breast milk compartment; $Q_{PQ}$, apparent intercompartmental clearance of primaquine from plasma to breast milk; $VM_{PQ}$, apparent volume of distribution of primaquine in breast milk; $PC_{CPQ}$, fraction of carboxyprimaquine that is freely distributed to breast milk compartment; $Q_{CPQ}$, apparent intercompartmental clearance of carboxyprimaquine from plasma to breast milk; $VM_{CPQ}$, apparent volume of distribution of carboxyprimaquine in breast milk; $MTINF_{PQ}$, first-order transfer rate from primaquine in breast milk compartment to infant primaquine dose compartment; $MTINF_{CPQ}$, first-order transfer rate from carboxyprimaquine in breast milk compartment to infant carboxyprimaquine dose compartment; $SQW_{milk-to-infant}$, the square-wave function regulating the breastfeeding pattern from breast milk compartments to infant dose compartments; $SQW_{venous-to-milk}$, the square-wave function regulating the transfer process from central compartments of primaquine and carboxyprimaquine to breast milk compartments; $k_{tr-INF}$, transit absorption rate constant in the infant; $CL_{PQINF}$, apparent elimination clearance of primaquine in infant; $V_{PQINF}$, apparent central volume of distribution of primaquine in infant; $CL_{CPQINF}$, apparent elimination clearance of carboxyprimaquine in infant; $V_{CPQINF}$, apparent central volume of distribution of carboxyprimaquine in infant.

at 0.348, interpreted as 65.2% lower breast milk concentrations compared to venous concentrations. For carboxyprimaquine, the $CFM_{CPQ}$ was estimated at 0.00877, interpreted as 99.1% lower breast milk concentrations compared to venous concentrations.

In the second approach, separate breast milk compartments for primaquine and carboxyprimaquine were introduced to the model. The transfer rates from the central compartments of primaquine and carboxyprimaquine to their breast milk compartments were estimated and assumed to be equal ($Q/F_{PQ} = Q/F_{CPQ}$). The volume of breast milk compartments of primaquine and carboxyprimaquine, derived from Eq. 1, using the average feeding frequency of ~10 feeds/day observed in the current study population ranged from 0.062 to 0.162 L. Additionally, the fraction of drug distributed from the central compartments of primaquine ($PC_{PQ}$) and carboxyprimaquine ($PC_{CPQ}$) to their breast milk compartment were estimated at 0.376 and 0.00889, respectively.

The schematic of the final pharmacokinetic model of primaquine and carboxyprimaquine using venous, capillary, and breast milk data is presented in Fig. 1. The pharmacokinetic parameters from the final pharmacokinetic model in breastfeeding women are summarised in Table 2. The goodness-of-fits (see Supplementary Fig. 1) showed no systematic bias. The prediction-corrected visual predictive checks in Fig. 2 showed a good predictive performance of the model. The predicted median total infant daily dose ranged from 1.18 to 5.09 µg/kg for the different dosing regimens evaluated in the mother, and the predicted median relative infant dose of primaquine ranged from 0.472 to 0.509% (i.e., about 200 times lower than the mother's weight-adjusted dose) as summarised in Table 3.

**Predicting infant concentrations**

The final population pharmacokinetic model of primaquine and carboxyprimaquine was linked to the infant model to construct the mechanistic mother-to-infant model (Fig. 1). All clearance and volume parameters in an infant were scaled from the values estimated in their mother using infant body weight and thereafter fixed. The maturation effect of monoamine oxidase A (MAO-A) enzyme was incorporated into the primaquine clearance in infants using infant age to account for the lower enzyme activity compared to adults. A postmenstrual age at which the clearance in infants was 50% of the mature clearance ($TM_{50}$) of 7.6 months was derived from the literature[26] to reflect 55% activity at birth and this value was implemented in the model. The absorption model for the infants was implemented as a 2-compartment transit model with a mean transit time fixed to 0.706 h[13].

In order to mimic the observed feeding pattern of 10 breastfeeds per day, the square-wave functions were adjusted with a breastfeeding cycle set to 2.4 h between feeds, resulting in a 24-minute feeding window followed by a 2-h break between feeds. The square-wave functions were multiplied to the transfer rate from the central compartments of primaquine and carboxyprimaquine to breast milk compartments ($Q_{PQ}$ and $Q_{CPQ}$), as well as the transfer rate from breast milk compartments to infant dose compartments ($MTINF_{PQ}$ and $MTINF_{CPQ}$). The $MTINF_{PQ}$ and $MTINF_{CPQ}$ were fixed to 100 to ensure that >95% of the amount in the breast milk compartment was transferred during the feeding window. This mother-to-infant model was then used for simulations.

The simulated infant concentration-time profiles (n = 1000), with the study dose administered to mothers, were overlaid with the available observed infant concentrations (Fig. 3). The data below the lower limit of quantification (LLOQ) were replaced with half of the LLOQ values, specifically 0.912 ng/mL for primaquine and 3.90 ng/mL for carboxyprimaquine, to allow for visualisation of data. The prediction intervals from the simulations closely aligned with the observed primaquine and carboxyprimaquine data, showing median concentrations below the LLOQ for both observed and predicted infant primaquine, and approximately 70% below the LLOQ for both observed and predicted carboxyprimaquine concentrations.

The sensitivity analysis of breastfeeding pattern, in which the number of feeds and the breast milk volume per feed were varied,

**Table 2 | Pharmacokinetic parameter estimates from the simultaneous population pharmacokinetic model of primaquine-carboxyprimaquine in breastfeeding women using plasma, capillary, and breast milk concentrations**

| Parameter | Population estimates [a] (%RSE) [b] | 95%CI [b] | IIV [a] [%CV] (%RSE) [b] | 95%CI [b] |
|---|---|---|---|---|
| F | 1 fixed | - | IIV: 15.7 (10.2) IOV: 20.5 (9.29) | IIV: 12.7–18.9 IOV: 16.8–24.9 |
| MTT (h) | 1.44 (9.43) | 1.19–1.77 | IIV: 20.5 (13.3) IOV: 56.8 (8.62) | IIV: 15.5–26.4 IOV: 48.1–69.3 |
| $F_M$ (%) | 0.282 (8.14) | 0.239–0.330 | 42.1 (11.2) | 32.6–51.7 |
| $CL/F_{PQ}$ (L/h) | 17.1 (5.85) | 15.1–18.9 | 15.1 (13.1) | 11.1–19.1 |
| $V/F_{PQ}$ (L) | 131 (7.26) | 114–151 | 19.2 (19.0) | 14.6–28.0 |
| $CL/F_{CPQ}$ (L/h) | 0.967 (7.56) | 0.825–1.12 | 26.7 (12.0) | 20.5–33.4 |
| $V/F_{CPQ}$ (L) | 22.7 (6.82) | 19.8–26.0 | 17.7 (15.9) | 12.8–23.6 |
| $CF_{PQ}$ | 0.898 (2.69) | 0.851–0.943 | - | - |
| $CF_{CPQ}$ | 1.06 (1.25) | 1.03–1.08 | - | - |
| $Q/F_{PQ}$ (L/h) [c] | 0.400 (19.4) | 0.282–0.582 | 89.7 (8.76) | 69.3–110 |
| $Q/F_{CPQ}$ (L/h) [c] | 0.400 (19.4) | 0.282–0.582 | 89.7 (8.76) | 69.3–110 |
| $V_M/F_{PQ}$ (L) | Calculated using infant body weight [d] | - | - | - |
| $V_M/F_{CPQ}$ (L) | Calculated using infant body weight [d] | - | - | - |
| $PC_{PQ}$ | 0.376 (3.70) | 0.350–0.405 | - | - |
| $PC_{CPQ}$ | 0.00889 (3.13) | 0.00834–0.00945 | - | - |
| $\sigma_{PQ-venous}$ | 0.102 (3.93) | 0.0863–0.118 | - | - |
| $\sigma_{CPQ-venous}$ | 0.0198 (4.01) | 0.0169–0.0234 | - | - |
| $\sigma_{PQ-capillary}$ | 0.0570 (5.95) | 0.0453–0.0733 | | |
| $\sigma_{CPQ-capillary}$ | 0.0115 (6.05) | 0.00902–0.0145 | | |
| $\sigma_{PQ-breast\ milk}$ | 0.156 (6.47) | 0.123–0.203 | - | - |
| $\sigma_{CPQ-breast\ milk}$ | 0.0911 (5.09) | 0.0733–0.110 | - | - |
| Secondary pharmacokinetic parameters of primaquine and carboxyprimaquine | | | | |
| | Primaquine | Carboxyprimaquine | | |
| Plasma | Median (min-max) | Median (min-max) | | |
| $T_{MAX}$ (h) | 2.80 (1.29–5.87) | 6.78 (4.71–12.3) | | |
| $C_{MAX}$ (ng/mL) | 120 (44.8–184) | 1255 (822–2194) | | |
| $AUC_{0-336h}$ (µg × h/mL) | 14.3 (1.41–31.4) | 349 (40.2–681) | | |
| Half-life (h) | 4.95 (4.05–9.78) | 17.2 (10.0–27.4) | | |
| Breast milk | | | | |
| $T_{MAX}$ (h) | 3.43 (1.77–6.22) | 7.67 (5.01–12.5) | | |
| $C_{MAX}$ (ng/mL) | 44.7 (16.6–69.0) | 11.2 (7.26–19.5) | | |
| $AUC_{0-336h}$ (µg × h/mL) | 5.38 (0.530–11.8) | 3.10 (0.357–6.05) | | |

[a] Population mean values, inter-individual variability (IIV) and inter-occasion variability (IOV) were estimated by NONMEM. The coefficient of variation (%CV) for IIV and IOV was calculated as $100 \times \sqrt{\exp(\text{estimate}) - 1}$.

[b] Relative standard error (%RSE) and 95 % confidence interval (95%CI) were assessed by sampling importance resampling (SIR).

[c] These values were set to be equal.

[d] Volume of breast milk compartment was calculated as $V_M = \frac{0.15 (L/kg/day) \times \text{infant body weight} (kg)}{\text{number of feeds} (/day)}$.

PQ, primaquine; CPQ, carboxyprimaquine; F, relative bioavailability; MTT, mean transit absorption time; $F_M$, fraction of primaquine converted to carboxyprimaquine during first-pass metabolism; CL, apparent elimination clearance; V, apparent volume of distribution of the central compartment; CF, conversion factor between venous and capillary drug concentrations; Q, apparent inter-compartmental clearance between central and breast milk compartment; $V_M$, apparent volume of distribution of breast milk compartment; PC, fraction of drug distributed from central compartment to breast milk compartment; σ, variance of the residual variability incorporated as an additive error on the logarithmic scale; $T_{MAX}$, time to reach maximum concentration; $C_{MAX}$, maximum concentration; $AUC_{0-336h}$, area under the concentration-time curve from time 0 to 336 h.

showed no major impact on the predicted concentration-time profile, the maximum concentration ($C_{MAX}$), or area under the concentration-time curve (AUC) in infants for both primaquine and carboxyprimaquine (see Supplementary Figs. 2–4). Exposure to both primaquine and carboxyprimaquine was comparable across all feeding frequencies. Slightly higher variability in the predicted $C_{MAX}$ was observed with scenarios involving larger breast milk volume per feed. But the overall predicted exposures remained significantly below maternal exposure (approximately 100-fold lower exposures), even in an extreme hypothetical example involving >500 ml of breast milk per feed. For the sensitivity analysis of MAO-A enzyme maturation, varying

enzyme activity at birth resulted in a higher predicted primaquine concentration in infants. However, the predicted concentration remained below the LLOQ, even in the most extreme scenario of slow enzyme maturation, where the enzyme had an activity of 12.5% at birth and reached 90% activity at 45 years old (see Supplementary Fig. 5).

Since the mother-to-infant model demonstrated adequate predictions of infant concentrations in line with observed data, this model was carried forward and used for different dosing scenarios simulations. The simulations predicting exposure in infants were summarised in three different ways; in Fig. 4 simulated concentration-time profiles of primaquine are presented, in Fig. 5 $C_{MAX}$ of primaquine are plotted

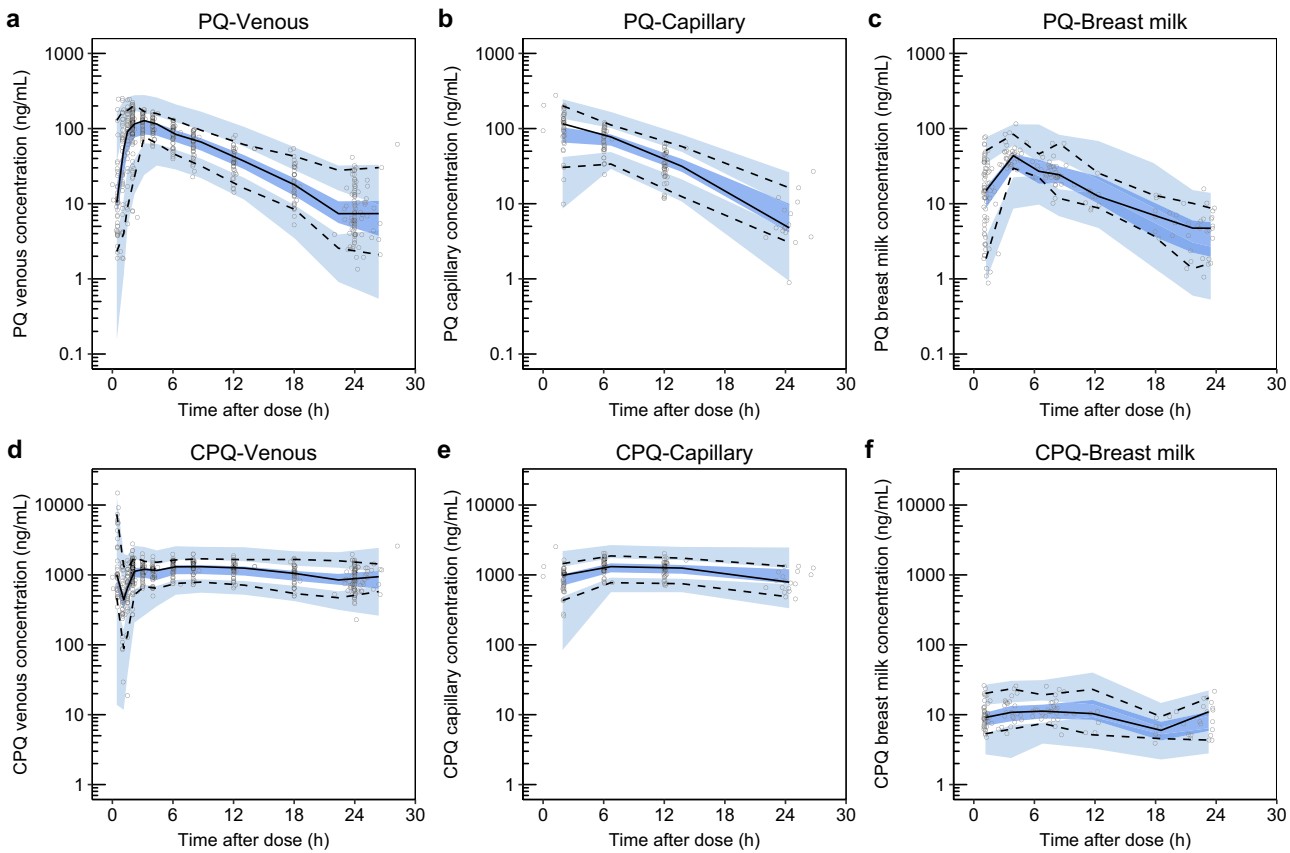

**Fig. 2 | Prediction-corrected visual predictive checks (n = 1000) of the final population pharmacokinetic model of primaquine (PQ) and carboxyprimaquine (CPQ) in breastfeeding women. a** Primaquine concentration in venous plasma. **b** Primaquine concentration in capillary plasma. **c** Primaquine concentration in breast milk. **d** Carboxyprimaquine concentration in venous plasma. **e** Carboxyprimaquine concentration capillary plasma.

**f** Carboxyprimaquine concentration in breast milk. The open circles represent the observed concentrations. Solid black lines represent the median of the observations, and dashed black lines represent the 5th and 95th percentiles of the observations. The shaded areas represent the 95% confidence intervals of each simulated percentile.

**Table 3 | Total infant daily dose and relative infant dose of primaquine simulated from the final pharmacokinetic model in the breastfeeding mother**

| Parameter | Standard dose 0.5 mg base/kg OD for 14 days | High dose 1 mg base/kg OD for 7 days | High dose 0.5 mg base/kg BID for 7 days | Single low dose 0.25 mg base/kg single dose |
|---|---|---|---|---|
| Maternal daily dose (mg/kg) | 0.5 | 1 | 1 | 0.25 |
| Total infant daily dose (µg/kg) | 2.54 (0.781–7.48) | 5.09 (1.41–14.0) | 5.09 (1.56–14.9) | 1.18 (0.323–4.47) |
| Milk to plasma ratio, (AUC$_{milk}$/AUC$_{venous}$) | 0.376 (0.375–0.377) | 0.376 (0.375–0.377) | 0.376 (0.375–0.377) | 0.375 (0.337–0.376) |
| Relative infant dose (%) | 0.509 (0.156–1.50) | 0.509 (0.141–1.40) | 0.509 (0.156–1.49) | 0.472 (0.129–1.79) |

Values reported as median (min-max). OD, once daily dosing; BID, twice daily dosing; AUC, total area under the concentration-time curve.

against infant age, and in Fig. 6 AUC of primaquine are plotted against infant age. The same type of plots for carboxyprimaquine were presented in supplementary (see Supplementary Figs. 6–8). For $C_{MAX}$ and AUC, the simulated exposures in infants were also compared to the well-tolerated threshold in mothers receiving a single low dose of primaquine (0.25 mg base/kg). Overall, the median simulated concentration-time curve of primaquine concentrations in infant were below LLOQ in all dosing scenarios. The predicted $C_{MAX}$ and AUC plotted against infant age showed the same trend for all dosing scenarios, the predicted $C_{MAX}$ and AUC slightly decreased with increasing infant age. The results showed that all median predicted $C_{MAX}$ and AUC

in infants were > 100 times lower compared to the median $C_{MAX}$ and AUC in the mother for all simulated dosing scenarios. Moreover, when comparing the median primaquine exposure in infants to the median primaquine exposure in mothers receiving a single low dose ($C_{MAX}$ = 55.0 ng/mL, AUC = 512 ng × h/mL), all of the simulated $C_{MAX}$ and AUC of primaquine in infants were well-below this threshold in the dosing scenarios evaluated.

## Discussion

In the present work a mechanistic population pharmacokinetic model of primaquine and carboxyprimaquine in breastfeeding women was

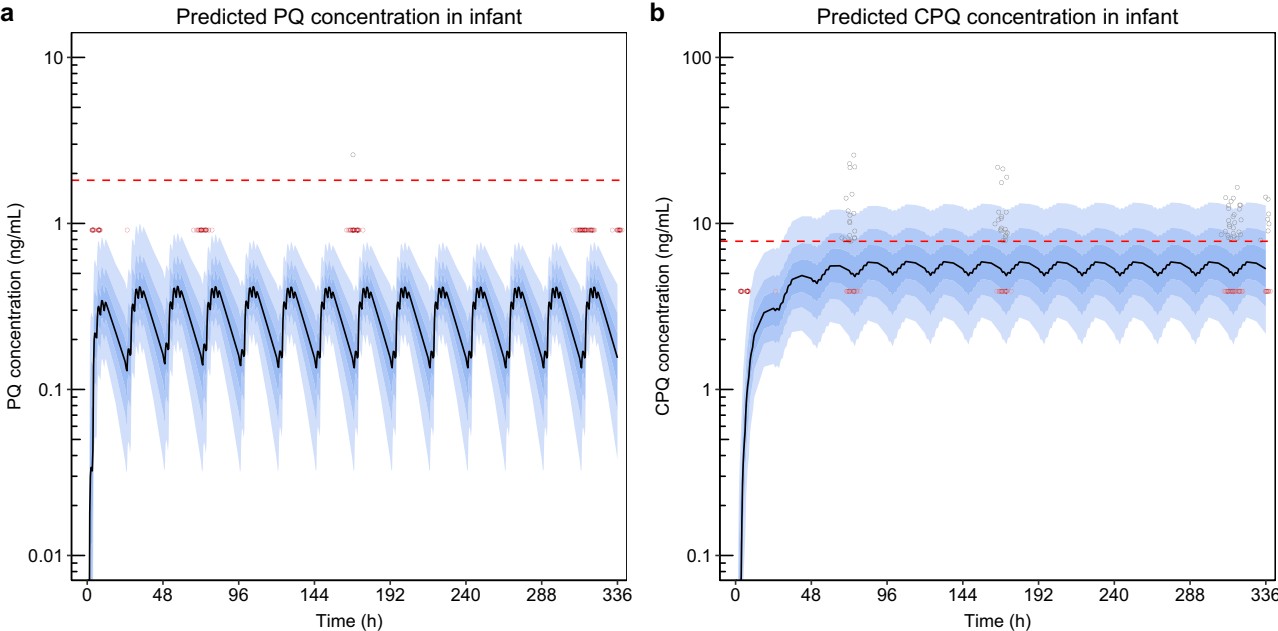

**Fig. 3 | The simulations from the mother-to-infant model predicting the concentration-time profiles in infants (n = 1000). a** Predicted primaquine (PQ) concentration in infants. **b** Predicted carboxyprimaquine (CPQ) concentration in infants. The open circles represent the observed concentrations in infants. The dashed red lines represent the median lower limit of quantification (LLOQ) of primaquine (PQ, 1.82 ng/mL) and carboxyprimaquine (CPQ, 7.81 ng/mL) reported in infant capillary samples. Observations that were measured below the LLOQ were substituted by LLOQ/2 and presented in the plots in red colour. The solid black lines represent the median of the simulations. The shaded areas represent the 95% prediction intervals of the simulations.

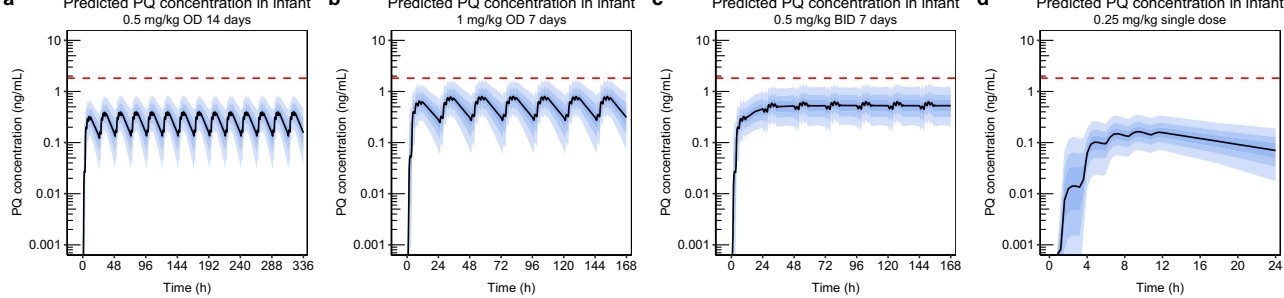

**Fig. 4 | Simulated concentration-time profile of primaquine from the mother-to-infant model using different dosing regimens (n = 1000). a** 0.5 mg base/kg once daily for 14 days. **b** 1 mg base/kg once daily for 7 days. **c** 0.5 mg base/kg twice daily for 7 days. **d** 0.25 mg base/kg single dose. The solid black lines represent the median of the simulations. The shaded areas represent the 95% prediction intervals of the simulations. The dashed red lines represent the median lower limit of quantification of primaquine (PQ, 1.82 ng/mL) reported in infant capillary samples.

successfully developed to describe venous, capillary, and breast milk concentrations. This model allowed predictive estimates of infant drug exposure at different doses of primaquine, in all cases predicting low infant exposure, compatible with safe use even at higher doses administered to mothers.

The final pharmacokinetic parameters estimated in breastfeeding women are similar to previous pharmacokinetic reports in non-pregnant adults[10–12]. The model estimated that primaquine was distributed to the breast milk compartment resulting in primaquine concentrations equivalent to 37.6% (95%CI: 35.0 to 40.5%) of that seen in plasma, and that only a very small fraction of carboxyprimaquine was distributed to the breast milk compartment (0.889% (95%CI: 0.834 to 0.945%) of that seen in plasma). The estimate for primaquine was close to the day-13 milk to plasma ratio of 0.37 (range: 0.24-0.61) reported in a previous analysis[25].

Breast milk may have different compositions of fat and a varying pH value during different breastfeeding periods. This could impact the distribution of primaquine into the breast milk and thus affect the estimation of milk to plasma ratio. A recently published physiologically-based pharmacokinetic (PBPK) model of primaquine[27] showed that the fat content had negligible impact on the predicted milk to plasma ratio. However, changing pH from 7.6 in colostrum to 7.2 in mature milk resulted in ~2-fold increase in the predicted milk to plasma ratio. The breast milk data collected in the current study was mature milk, which have smaller amounts of fat and lower pH value compared to colostrum. The impact of different breast milk composition in the early breastfeeding period is currently assessed in an ongoing clinical study (ClinicalTrials.gov Identifier: NCT04984759). The study will examine primaquine concentration in colostrum, and a revised pharmacokinetic model for evaluating drug exposure in these women could be developed when data become available.

The developed mother-to-infant model predicted the infant concentrations of primaquine to be below the LLOQ mirroring what was seen in the observed data. Similarly, the concentrations of

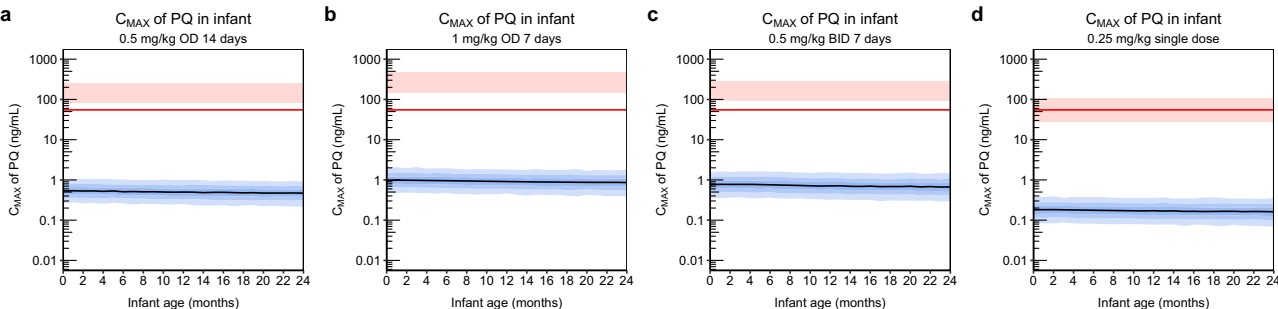

**Fig. 5 | Simulated $C_{MAX}$ of primaquine from the mother-to-infant model using different dosing regimens (n = 1000). a** 0.5 mg base/kg once daily for 14 days. **b** 1 mg base/kg once daily for 7 days. **c** 0.5 mg base/kg twice daily for 7 days. **d** 0.25 mg base/kg single dose. The solid black lines represent the median of the simulations in infants. The blue shaded areas represent the 95% prediction intervals of the simulations in infants. The red shaded areas represent the 95% prediction interval of the simulated $C_{MAX}$ in the mothers with 60 kg body weight of each dosing scenario. The red solid lines represent the median $C_{MAX}$ (55.0 ng/mL) of the mother receiving single low dose of primaquine (0.25 mg base/kg), a conservative reference for primaquine concentration known not to cause significant haemolysis in G6PD deficient individuals.

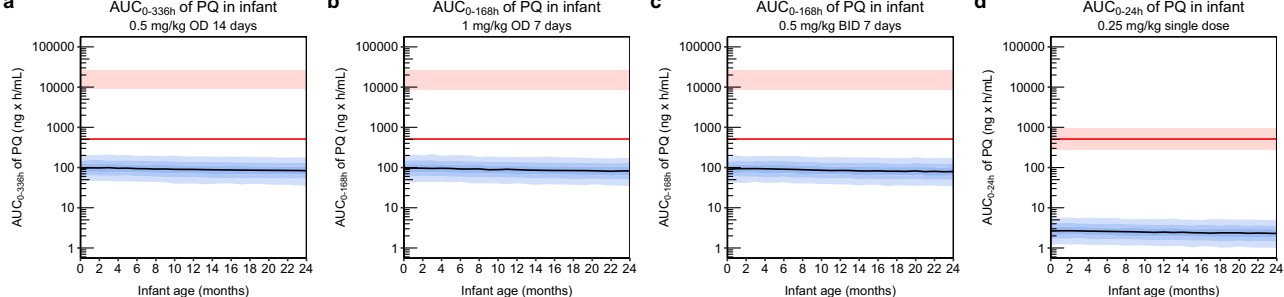

**Fig. 6 | Simulated AUC of primaquine from the mother-to-infant model using different dosing regimens (n = 1000). a** 0.5 mg base/kg once daily for 14 days. **b** 1 mg base/kg once daily for 7 days. **c** 0.5 mg base/kg twice daily for 7 days. **d** 0.25 mg base/kg single dose. The solid black lines represent the median of the simulations in infants. The blue shaded areas represent the 95% prediction intervals of the simulations in infants. The red shaded areas represent the 95% prediction interval of the simulated AUC in the mothers with 60 kg body weight. The red solid lines represent the median $AUC_{0-24h}$ (512 ng × h/mL) of the mother receiving single low dose of primaquine (0.25 mg base/kg), a reference for a primaquine exposure known not to cause significant haemolysis in G6PD deficient individuals.

carboxyprimaquine in infants were predicted to be close to the available observed concentrations, with approximately 70% predicted to be below LLOQ.

The volume of the breast milk intake per feed varies among infants, with the age of the infant being the primary factor influencing the feeding volume[28]. In this model, the volume of breast milk intake by the infant was varied based on infant body weight, and the number of feeds per day were assumed to be equal to the average values seen in the study population, without adjustment for age. However, the results from sensitivity analysis showed that differences in feeding volume resulted in similar drug exposure in infants.

Enzyme activity could have an impact on primaquine exposure in infants. However, the information about MAO-A enzyme activity in neonates and infants is scarce. An enzyme activity of 55% at birth compared to adults, used in the current analysis, was based on the only available publication describing human data. The sensitivity analysis of varying MAO-A enzyme activity at birth was performed to evaluate the impact of MAO-A enzyme maturation on the predicted primaquine concentration in infants. The model predicted higher primaquine concentration in infants with lower enzyme activity at birth. However, the predicted primaquine concentration in infants still remained below the LLOQ in all scenarios. The PBPK model of primaquine[27] showed similar predicted primaquine concentration of approximately 1 ng/mL in neonates when implementing a slow ontogeny of the MAO-A enzyme.

Primaquine predominantly binds to plasma α1-acid glycoprotein (AAG). The AAG levels in neonates are generally lower compared to adults, which gradually increase and reach adult levels during adolescence[29]. The impact of changing AAG levels in healthy infants aged 1 month to 2 years (the age range of infants in this study) is expected to be limited, and since the AAG level was unavailable, it was not examined in the current analysis. The sensitivity analysis from the PBPK model revealed that a four-fold increase in AAG, which was observed in children with malaria and severe malnutrition, increased the relative exposure in infants compared to mothers from 0.12% to 0.35%. However, it should be noted that the infants evaluated here did not have malaria or any other co-morbidities that would result in an increased AAG level.

As 5-hydroxyprimaquine, the metabolite that is likely involved in primaquine-induced haemolysis, is metabolised by CYP2D6. Enzyme immaturity in infants and genetic polymorphisms of CYP2D6 enzyme could impact the production of 5-hydroxyprimaquine and thus the level of haemolysis. However, this is unlikely to uncover higher levels of toxicity than predicted here, as infants have lower hepatic enzymatic activity than adults[19,21,22,30], leading to lower production of haemolytic metabolites compared to adults or older children. Additionally, due to the reactive and unstable nature of 5-hydroxyprimaquine, it is highly unlikely that such a compound could be distributed and accumulate in breast milk. To enhance the characterization of the relationship between primaquine and its haemolytic effects, a pharmacokinetic/pharmacodynamic models with the measurement of a more stable metabolite of 5-hydroxyprimaquine (5,6-orthoquinone)[19], could be conducted across various CYP2D6 polymorphisms.

In the previous clinical study, the safety and adverse events after receiving standard dose of primaquine (0.5 mg base/kg once a day for

14 days) were monitored in healthy breastfeeding women and their breastfed infants. Primaquine was well tolerated and no serious adverse events were reported in this population[25]. A study conducted in villages along the Thai-Myanmar border has shown that single low dose of primaquine (0.25 mg base/kg) does not cause clinically significant haemolysis in G6PD-deficient individuals[31]. The results from the current study showed that the median infant exposure ($C_{MAX}$ and AUC) of both primaquine and carboxyprimaquine were > 100 times lower than the median exposure in their mother for all simulated dosing scenarios. Moreover, all of the simulated dosing scenarios resulted in $C_{MAX}$ and AUC of primaquine in infants well-below that of the reference threshold observed in mothers receiving a single low dose of primaquine. Additional simulations were conducted to investigate whether the 8-weekly dosing regimen (0.75 mg/kg once weekly for 8 weeks), typically administered to patients with G6PD deficiency, would result in even lower exposure in infants. The results suggested that the predicted AUC in infants was slightly lower in the 8-weekly dosing regimen. However, it was generally comparable to the predicted exposure in the standard dosing regimen and remained below the LLOQ. These results support recommendations to give primaquine at therapeutic doses to breastfeeding mothers regardless of infant G6PD status, as it is highly unlikely that with this low primaquine exposure in infants would cause haemolysis.

The analysis utilised a relatively low number of 21 mother-infant pairs. However, the pharmacokinetic sampling schedule was well-designed and the model developed for the mothers described the data accurately in all sample matrices, and the uncertainty assessment of all estimated pharmacokinetic parameters showed that the model was robust (RSE < 20%). The ongoing study in neonates (i.e., data in colostrum) could fill the information gap in the early period of breast feeding.

The lack of measurable primaquine concentrations in the clinical infant samples is reassuring, but also the main limitation of the study and the reason why a full model describing the pharmacokinetics of primaquine and carboxyprimaquine in both mother and infant could not be developed. To the best of our knowledge, there is no established pharmacokinetic model in infants that could be connected to the developed pharmacokinetic model in the mother. Nonetheless, simulations using pharmacokinetic parameters from mothers scaled by infant body weight, accounting for enzyme maturation in infants, and incorporating a rapid absorption described in children, produced predictions that align with the observed concentrations.

The results from the current analysis were also comparable to those presented in a recently developed PBPK model of primaquine[27]. In the PBPK model, the median predicted maximum primaquine concentration in breastfed infants aged > 28 days was 0.14 ng/mL. However, the PBPK model for infants did not account for MAO-A enzyme maturation. In the current analysis, we assumed 55% enzyme activity at full-term birth, resulting in a median predicted maximum primaquine concentration in infants of 0.31 ng/mL which decreased to 0.24 ng/mL when excluding the maturation effect from the model. These differences may be partly explained by the different modelling approaches, wherein the PBPK model assumes more complex tissue compartments. Additionally, the variation in the milk to plasma ratio used in the simulation (0.34 in the PBPK model vs. 0.38 in the current analysis) could also contribute to this discrepancy.

Although direct comparison of model predictions to actual concentrations is not feasible when only LLOQ data are available, the developed mother-to-infant model predicted median primaquine concentrations to be below the LLOQ. This demonstrate that the model's predictions of primaquine concentrations in breastfed infants is close to the actual true values with an absolute maximum error of 1.82 ng/mL (LLOQ). This discrepancy is expected to be negligible when compared to concentrations observed in adults. Additional data in neonates and infants directly treated with primaquine are required to enhance the characterization of the pharmacokinetic properties in this population.

In conclusion, the results from the population pharmacokinetic modelling and simulations in this study suggest very low infant exposures to primaquine and carboxyprimaquine compared to maternal exposures. Such exposures are very unlikely to pose any haemolytic risk even in the most severe variants of G6PD deficiency. There is, therefore, no pharmacological reason to prevent breastfeeding women from taking primaquine. This supports the use of primaquine for radical cure in breastfeeding women with infants at least 28 days old and a history of *P. vivax* malaria. Therefore, we propose an amendment to the current treatment guidelines to include breastfeeding women with infants at least 28 days old in the radical cure of *P. vivax* malaria.

## Methods
### Clinical study
The data in this analysis were obtained from a clinical trial conducted at three clinics at Shoklo Malaria Research Unit. The clinics were located at the Thai-Myanmar border and the study was conducted between 11 November 2012 and 24 June 2014. Lactating women, at least 18 years of age, with a history of *Plasmodium vivax* infection, with no previous primaquine treatment for radical cure, with a currently breastfeeding infant at least 28 days of age, and provide informed consent were recruited to the study. Consenting mothers and their infants underwent both complete blood count and G6PD testing. The G6PD testing included a rapid G6PD fluorescent spot test (R&D Diagnostic, Greece) as well as G6PD genotyping through polymerase chain reaction–restriction fragment length polymorphism, specifically targeting the prevalent local variant (Mahidol). Any detected abnormalities in either the mother or the infant resulted in their exclusion from the study. The mothers were also screened for malaria smear, biochemistry, blood group, and haemoglobin typing. (Clinical Trials Registration: NCT01780753)[25].

### Ethics
Ethical approval for the study was acquired from three different bodies. The Ethics Committee of the Faculty of Tropical Medicine, Mahidol University, Bangkok (TMEC 12–036); the Oxford Tropical Research Ethics Committee (OXTREC 28-12); and the Tak Community Advisory board[32].

### Dosing and Sample collection
Primaquine 0.5 mg base/kg was given once daily for 14 days under non-fasting conditions as a directly observed oral therapy to each participating woman. Dense pharmacokinetic samples were collected on days 0 and 13, with sparse samples being collected on days 3 and 7. Venous samples were collected on hours 0, 0.5, 1, 1.5, 2, 3, 4, 6, 8, 12, 18, and 24 after the dose on day 0 and day 13. On days 3 and 7, samples were collected pre-dose and 2 h after dosing. Capillary samples were collected at 0, 2, 6, and 12 h after the dose on day 0 and 13. Breast milk was collected through manual expression in the following time windows: 1–3, 3–7, 7–12, 12–24 h after the dose on day 0 and 13. In addition to these samples, breast milk samples were also collected one time between 1 and 3 h on day 3 and 7. Capillary blood samples were also collected in the infants at the following timepoints: 0, 2, and 6 h after the first breastfeed following maternal primaquine ingestion on day 0, at 0 and 2 h after the first breastfeed on day 3 and 7, and at 0, 2, 6, and 24 h after the first breastfeed on day 13. All blood samples were centrifuged on-site, and the resulting plasma aliquots were immediately frozen. For breast milk samples, the total volume of expressed milk was measured, and a 2-ml aliquot was taken and immediately frozen. Samples were kept on ice until it could be stored in -80 °C and shipped to the Department of Clinical Pharmacology at the Mahidol Oxford Tropical Medicine Research Unit, Bangkok for drug quantification.

## Drug quantification

Collected samples (venous plasma from the mothers, capillary plasma from the mother and infants, and breast milk) were analysed for both primaquine and carboxyprimaquine concentrations using a high-performance liquid chromatography coupled with a mass-spectrometer[33]. Analytes were separated using reverse-phase high-performance liquid chromatography and quantified by electrospray ionisation in the positive mode with multi-reaction monitoring mass spectrometry detection. The lower limit of quantification (LLOQ) was 1.14 ng/ml for primaquine and 4.88 ng/ml for carboxyprimaquine in all sample matrices. The relative standard error was below 10% for all drug measurements.

## Pharmacokinetics in breastfeeding women

The natural logarithm of observed molar primaquine and carboxyprimaquine concentrations was analysed using non-linear mixed effect modelling as implemented in NONMEM version 7.4 (Icon Development Solution, Ellicott City, MD) using the first-order conditional estimation method with interactions. Automations, model evaluations, model diagnostics and model support were carried out using the R-package Xpose version 4.0, Perl-speaks-NONMEM version 4.8.0, as well as Pirana version 2.9.9[34–36]. The OFV was assumed to be $\chi^2$ distributed and a drop of >3.84, >6.63, or >10.83 were considered statistically significant at a $P$ value of <0.05, <0.01, and <0.001, respectively. Pharmacokinetic parameters were assumed to be log-normally distributed, and IIV was implemented with an exponential function. The IOV, was also implemented with an exponential function to investigate the random variability between sampling occasions i.e., day 0, day 3, day 7, and day 13 and was evaluated on all absorption parameters.

Primaquine and carboxyprimaquine were fitted simultaneously and all possible combinations of one- and two-compartment disposition models were evaluated for the two compounds to describe the drug and metabolite distribution. Different models to describe the absorption of primaquine were evaluated; both a first-order absorption as well as a transit compartment model[37]. Both estimating $k_a$ and $k_{tr}$ separately as well as assuming them to be equal were evaluated. In addition, direct metabolism from the last transit compartment to the central compartment of carboxyprimaquine (i.e., first-pass metabolism) was evaluated[11]. Relative oral bioavailability of primaquine was evaluated by fixing the relative bioavailability to 100% for the population and estimating an IIV in the same parameter. As both venous and capillary concentrations were available, a proportional conversion factor between venous and capillary prediction was estimated.

Covariate effects were evaluated using both venous and capillary data. Body weights were implemented as an allometric covariate on all clearance and volume parameters with an exponent of 0.75 for clearance parameters and 1 for volume parameters[38,39]. Other admission covariates including age (years), haemoglobin (mg/dL), total bilirubin (mg/dL), aspartate aminotransferase (AST) (U/L), alanine aminotransferase (ALT) (U/L), alkaline phosphatase (ALP) (U/L), and smoking status were explored on all pharmacokinetic parameters using a stepwise covariate search method as implemented in PsN. The significance level was defined at a $P$ value of <0.05 for forward inclusion and <0.001 for backward elimination.

After the models for primaquine and carboxyprimaquine had been established using venous and capillary data, the breast milk data were introduced. Two separate approaches were tried. In the first approach, a conversion factor of the venous plasma concentrations in the central compartment for both primaquine and carboxyprimaquine were estimated to fit the breast milk concentration-time data. In the second approach, the distribution of primaquine and carboxyprimaquine from the central compartments into separate breast milk compartments (one compartment for primaquine and one compartment for carboxyprimaquine) were estimated to fit the breast milk concentration-time data. For the second approach, it was assumed that the transfer rate from the central compartment to the breast milk compartment was the same for primaquine and carboxyprimaquine (Fig. 1). The volume of the breast milk compartment was calculated using the amount of milk ingested per feed, which was derived from the average amount of daily milk intake (150 ml/kg infant body weight)[28] and the average reported feeding frequency in the current study population, as shown in Eq. 1.

$$V_M = \frac{150(\text{mL/kg/day}) \times \text{infant body weight (kg)}}{\text{number of feeds (/day)}} \quad (1)$$

The $PC_{PQ}$ and $PC_{CPQ}$ described the distribution of primaquine and carboxyprimaquine into the breast milk compartments and were estimated. The final model was evaluated using basic goodness-of-fit criteria as well as simulation-based diagnostics in the form of a prediction-corrected visual predictive check[40]. Uncertainty in model parameter estimates was obtained using the iterative sampling importance resampling (SIR) procedure[41].

In order to obtain the total infant daily dose and relative infant dose of primaquine, simulations from the final model of the breast-feeding mother were performed using four different maternal primaquine dosing scenarios, i.e., 0.5 mg base/kg once daily for 14 days (standard dose), 0.5 mg base/kg twice daily for 7 days (high dose twice daily), 1.0 mg base/kg once daily for 7 days (high dose once daily), and 0.25 mg base/kg single dose (single low dose). The body weights of breastfeeding mothers were taken from the study population (35 to 81 kg) and were used in all simulations. The total infant daily dose and the relative infant dose of primaquine were calculated using Eqs. 2 & 3, respectively.

$$\text{Total infant daily dose} = \frac{AUC_{\text{milk}}(\mu g \times h/mL)}{AUC_{\text{venous}}(\mu g \times h/mL)} \times C_{ss,\text{average}}(\mu g/mL) \times 150(\text{mL/kg}) \quad (2)$$

$$\text{Relative infant dose (\%)} = \frac{\text{infant dose}(mg/kg)}{\text{maternal dose}(mg/kg)} \times 100 \quad (3)$$

Where $AUC_{\text{milk}}$ represents the area under the concentration-time curve of primaquine in breast milk from 0 h to 24 h after the last dose, $AUC_{\text{venous}}$ represents area under the concentration-time curve of primaquine in venous plasma from 0 h to 24 h after the last dose, and $C_{ss,\text{average}}$ represents the average concentration of primaquine in venous plasma at steady state.

## Predicting infant concentrations

Concentration-time data in infants were limited, as all capillary plasma primaquine concentrations in infants were below the LLOQ, except one sample that was measured at 2.59 ng/mL. In addition, 67.5% of carboxyprimaquine concentrations in infants were below the LLOQ, and so pharmacokinetic parameters could not be estimated in infants. Therefore, to be able to predict concentrations in infants, simulations were performed using a proposed mechanistic mother-to-infant model (Fig. 1). The infant model was assumed to be identical to the primaquine-carboxyprimaquine model developed in the mothers, but ignoring the first-pass metabolism of primaquine into carboxyprimaquine. The clearance and volume parameters from the mother were scaled allometrically using infant body weight. The maturation effect MAO-A enzyme in infants, responsible for converting primaquine to carboxyprimaquine was incorporated. It was assumed that the enzyme activity at full-term birth was 55% compared to adults[26].

This was implemented on infant's primaquine clearance using Eq. 4.

$$MF = \frac{PMA}{PMA + TM_{50}} \quad (4)$$

Where MF denotes the maturation effect used to multiply with the apparent clearance of primaquine in infants, PMA denotes postmenstrual age in months, assuming full-term gestation at 9.2 months (i.e., 40 weeks), and $TM_{50}$ is the PMA at which the clearance in infants is 50% of the mature clearance. As the maturation effect was derived from enzyme activity reported in the literature, the $TM_{50}$ reflecting 55% activity at birth was then calculated to be 7.6 months.

The mean transit absorption model in infant was fixed to a literature model reported in children, in order to account for more rapid absorption[13]. This absorption model was implemented both for primaquine and carboxyprimaquine in the infants, as they ingest both compounds through the breast milk. The transfer of primaquine and carboxyprimaquine to the infant was modelled as a fixed first-order transfer rate from the breast milk compartment of primaquine and carboxyprimaquine to the infant dose compartments. The $MTINF_{PQ}$ and $MTINF_{CPQ}$ was fixed to a very high value to ensure that > 95% of the drug amount in the breast milk compartment was transferred to the infant during the feeding window. In order to mimic the feeding pattern during the day, square-wave functions were integrated into the model and applied as multipliers to the transfer rates, which affected both the transfer rate from the central compartments of primaquine and carboxyprimaquine to breast milk compartments, as well as the transfer rate from breast milk compartments to infant dose compartments. The square-wave functions were configured to alternate between 1 and 0. This facilitated the activation of the transfer process from breast milk compartments during breastfeeding sessions, while deactivating it during the resting periods. Conversely, the central compartments of primaquine and carboxyprimaquine followed the opposite pattern. The activation and deactivation adjusted at specific time intervals according to the number of feeds per day. The square-wave functions were described using Eqs. 5–8.

$$S_1 = \sin\left(\frac{\left(\pi - \left(\frac{2\pi \times SH_1}{T_{CYCLE}}\right)\right)}{2}\right) \quad (5)$$

$$S_2 = \sin\left(\left(\frac{2\pi \times time}{T_{CYCLE}}\right) + \frac{\left(\pi - \left(\frac{2\pi \times SH_1}{T_{CYCLE}}\right) + SH_2\right)}{2}\right) \quad (6)$$

$$SQW_{milk-to-infant} = \left(\sqrt{\frac{\left((S_2 - S_1)^2 - (S_2 - S_1)\right)}{\left(2 \times \sqrt{(S_2 - S_1)^2}\right)}}\right) \quad (7)$$

$$SQW_{venous-to-milk} = (SQW_{milk-to-infant} - 1) \times (-1) \quad (8)$$

where $S_1$ denotes the first sine-wave function, $S_2$ denotes the second sine-wave function, $SH_1$ denotes time when the function remains in state 0 (h), $SH_2$ denotes time when the function changes from state 0 to 1 (h), $T_{CYCLE}$ denotes duration of breastfeeding cycle (24 h/number of feeds per day, h), and $SQW_{milk-to-infant}$ denotes the square-wave function regulating the breastfeeding pattern from breast milk compartments to infant dose compartments. $SQW_{venous-to-milk}$ denotes the square-wave function regulating the transfer process from central compartments of primaquine and carboxyprimaquine to breast milk compartments (opposite to $SQW_{milk-to-infant}$). $SH_1$ and $SH_2$ values were adjusted based on number of feeds per day and duration of feeding window. Simulated breast milk concentrations, using the developed

model with and without the square-wave function, were overlaid with observed concentrations to evaluate that no model misspecification occurred when the square-wave function was included (see Supplementary Fig. 9). Details on how the square-wave functions were implemented in the model are shown in the Supplementary Methods (NONMEM code). NONMEM model file was also deposited to Zenodo repository[42].

The infant concentration-time profiles of primaquine and carboxyprimaquine were then simulated using the developed mechanistic mother-to-infant model. The IIV was not incorporated into the infant's pharmacokinetic parameters. Therefore, the variation in predicted infant concentration within the mother-to-infant model is solely attributed to the variability in the mother's pharmacokinetic parameters. The 95% prediction interval from the simulation was overlaid with the available observed data in infants to compare and evaluate if the predictions from mother-to-infant model were close to the observed data. Moreover, sensitivity analyses were performed to investigate the impact of different feeding pattern (varying feeding frequencies and feeding volume) on the predicted exposure to primaquine and carboxyprimaquine in the infant. Impact of various MAO-A enzyme maturation scenarios on predicted primaquine concentration in infants were also evaluated. Details on sensitivity analyses of feeding pattern and MAO-A enzyme maturation are described in the Supplementary Methods.

The final mother-to-infant model was then used to simulate primaquine and carboxyprimaquine concentration-time profiles using four different primaquine dosing scenarios in lactating women i.e. 0.5 mg base/kg once daily for 14 days (standard dose), 0.5 mg base/kg twice daily for 7 days (high dose twice daily), 1.0 mg base/kg once daily for 7 days (high dose once daily), and 0.25 mg base/kg single dose (single low dose). A mother body weight of 60 kg was used in all simulations. The infant's age and body weight in the simulation dataset ranged from 0 to 24 months and 2 to 17 kg, respectively. These ranges were determined based on WHO's weight-for-age standard growth curves with z-scores from -3 to 3 for newborn up to 2 years of age. A total of 1000 simulations (50,000 virtual infants) were performed for each dosing scenario. The simulated concentration-time profiles, $C_{MAX}$, and AUC were plotted against infant age to show the concentration and exposure of each dosing scenario. To assess the risk of haemolysis in infants, the simulated exposures in infants were compared to the simulated primaquine exposure in mothers who received a single low dose of primaquine (0.25 mg base/kg). This exposure level in the mother ($C_{MAX}$ and AUC) serves as a threshold for plasma exposure, known not to cause significant haemolysis in G6PD-deficient individuals, as reported in a previous study[31].

### Reporting summary

Further information on research design is available in the Nature Portfolio Reporting Summary linked to this article.

## Data availability

The patient data are available under restricted access for ethical reason, access can be obtained by written application to the Data Access Committee at Mahidol Oxford Tropical Medicine Research Unit (datasharing@tropmedres.ac). Applications are commonly reviewed within 2 weeks.

## Code availability

The population pharmacokinetic model in the current analysis was developed using NONMEM, a standardised software package. NONMEM code described both mother and mother-to-infant model used in this study was provided in the supplementary materials and NONMEM model file was also deposited to Zenodo repository (https://doi.org/10.5281/zenodo.10925291).

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

## Acknowledgements

The authors would like to express gratitude to all the women who participated in this study and to the infants who were enroled. We are also thankful for all the staffs at the Shoklo Malaria Research Unit (SMRU). SMRU is part of the Mahidol Oxford Research Unit and funded by Wellcome Trust. This work was supported by grants from the Institutional Translational Partnership Award (WT-iTPA 2020) and the Wellcome Trust (220211). M.E.G.'s DPhil is supported by the Tropical Network Fund of University of Oxford. For the purpose of Open Access, the author has applied a CC BY public copyright license to any Author Accepted Manuscript version arising from this submission.

## Author contributions

N.J.W., N.P.J.D., F.N. J.T., and R.M. conceptualized the study. W.H. and J.T. developed the assay to measure primaquine and carboxyprimaquine in breast milk and plasma. M.E.G. and W.H. generate the data. T.W., R.M.H. and J.T. performed the population pharmacokinetic analysis. T.W., R.M.H. and M.E.G. wrote the first draft of the manuscript. All authors revised and approved the final manuscript.

## Competing interests

The authors declare no competing interests.
