## [Peer Review File · Nature Communications]

Population pharmacokinetic modelling of primaquine exposures in lactating women and breastfed infantsReviewers' Comments:

Reviewer #1:

Remarks to the Author:

Thank you for the opportunity to review this interesting and impactful manuscript. Although the drug concentrations used for the model have been previously published, the modeling approach is innovative in the antimalarial field and there are important policy implications. The manuscript is very well written and provides sufficient detail to reproduce the results.

Other comments:

Line 34: I understand the sentiment of this final line of the abstract, but it's making a statement beyond the scope of the data. Could remove or make a more targeted statement on how you'd like recommendations to change (e.g. remove breastfeeding women as a prohibited group for PQ from WHO guidelines).

Line 37 & 39: missing "it" before "as kills"

Line 58: Is the dose-dependent here referencing daily dose or cumulative dose over the course of treatment?

Line 330: Given the thorough justification of the model in the methods/results sections, streamlining the repeated results/model talk (lines 336-361) and getting to some of the clinical interpretation of the data (e.g. volume of breast milk, CYP2D6 polymorphism, safety PD thresholds) more quickly could engage a larger audience.

Line 367: Could the authors also explicitly comment on the impact of weekly dosing for G6PD deficient (e.g. if we would expect even lower infant exposure).

Line 391: Among the limitations, it would be helpful to comment on the low number of participants in the study, and whether the authors thing adding additional subjects would be necessary to support a change in WHO recommendations.

Reviewer #2:

Remarks to the Author:

The authors describe the development and application of a population PK model for primaquine (PQ) in lactating mothers and their infants. The manuscript addresses an important topic regarding the use of PQ in breastfeeding mothers. Although the manuscript is well written, several key points should be addressed including the limitations of the model for predictions in infants. Specific comments include:

1. In the methods, much of the text relates to the actual clinical study conducted by Gilder et al. (2018). Whilst I understand that some of the information is required for the population PK model development, it does seem excessive especially as the reference is cited in multiple places.
2. The main objective of the study was to develop "a mechanistic pharmacokinetic model, which can be used to predict the exposures of infants to primaquine". However, the model was developed using clinical data from lactating women and applied in infants where there are no clinical/verification data to support the use of the model in this vulnerable population. What confidence do we have that the population PK model can be applied to infants?
3. There is considerable uncertainty associated with factors affecting the PQ exposure in infants, including enzyme ontogenies and plasma protein binding (AAG). Whilst the former was accounted for, sensitivity analyses assuming various ontogenies for MAO were not performed. Furthermore, AAG can vary significantly in infants especially in a disease population and affect the exposure of PQ. However, this was not assessed.
4. The authors did consider that the IDD could change depending on the PQ dose given to mothers or the amount of milk consumed by the infant. However, the impact of pH or fat content which could affect the milk to plasma ratio (M/P) and ultimately the IDD, was not accounted for or discussed.
4. The authors did not refer to other mechanistic methods such as PBPK modelling which could be used to predict the exposures of PQ in breastfeeding infants. Indeed, there was a recent publication,

where PBPK was used for this very purpose (<https://doi.org/10.1002/psp4.13090>).

5. In the conclusion, it is stated that PQ treatment should be considered for mothers with infants aged at least 28 days. If the relevant ontogenies have been applied, why can't the findings be extrapolated to infants < 28 days.

Response to reviewers

Dear reviewers,

Thank you for reviewing our manuscript and providing insightful and valuable comments. We have modified the manuscript based on your comments. A detailed response to each of the comment is provided below. We greatly appreciate the editors and reviewers' time for providing these helpful comments and suggestions.

Yours sincerely,

Richard Hoglund

Reviewer #1 (Remarks to the Author):

Thank you for the opportunity to review this interesting and impactful manuscript. Although the drug concentrations used for the model have been previously published, the modeling approach is innovative in the antimalarial field and there are important policy implications. The manuscript is very well written and provides sufficient detail to reproduce the results.

Other comments:

Line 34: I understand the sentiment of this final line of the abstract, but it's making a statement beyond the scope of the data. Could remove or make a more targeted statement on how you'd like recommendations to change (e.g. remove breastfeeding women as a prohibited group for PQ from WHO guidelines).

Thank you for your recommendation. We have revised the final line to read "After the neonatal period, primaquine should not be restricted for breastfeeding women." (Line: 31-32)

Line 37 & 39: missing "it" before "as kills"

Thank you for pointing this out. We have revised the manuscript accordingly.

Line 58: Is the dose-dependent here referencing daily dose or cumulative dose over the course of treatment?

Primaquine-induced haemolysis depends on the daily dose administered, the length of primaquine administration and the severity of G6PD deficiency (1). A previous study (2) investigated haemolysis in volunteers after receiving various primaquine dosing regimens. The findings revealed that a primaquine dose of 30 mg once daily for 14 days (total dose 420 mg) resulted in a higher extent of haemolysis compared to a dose of 15 mg once daily for 14 days (total dose 210 mg). Conversely, haemolysis was markedly reduced with 60 mg once weekly for 8 weeks (total dose 480 mg) and 45 mg once weekly for 8 weeks (total dose 360 mg) compared to the daily dosing regimens. We have elaborated the sentence in the manuscript to make it clearer (Line: 54-56)

Line 330: Given the thorough justification of the model in the methods/results sections, streamlining the repeated results/model talk (lines 336-361) and getting to some of the clinical interpretation of the data (e.g.

volume of breast milk, CYP2D6 polymorphism, safety PD thresholds) more quickly could engage a larger audience.

Thank you for your recommendation. We have removed redundant sentences that duplicated the results in the discussion section. The procedure for calculating breast milk volume was initially outlined as text in the original manuscript. In the revised manuscript, we have explicitly included the equation used for this calculation in the method section (Line: 151). Additionally, we introduced the topic of CYP2D6 polymorphism early in the introduction section (Line: 60-62). Regarding the safety threshold, we have provided an explanation in the method section (Line: 242-245), specifying that we used the exposure in the mother who received a single low dose of primaquine (0.25 mg/kg) as a threshold known not to cause clinically significant haemolysis in G6PD-deficient individuals, as reported in the previous publication.

Line 367: Could the authors also explicitly comment on the impact of weekly dosing for G6PD deficient (e.g. if we would expect even lower infant exposure).

We conducted additional simulations to address this question. The simulation results showed that the predicted exposure in infants was comparable between the 8-weekly dosing regimen (0.75 mg/kg once weekly for 8 weeks) and the standard dosing regimen (0.5 mg/kg once daily for 14 days). Specifically, the C_{MAX} in the weekly dosing regimen was slightly higher compared to the standard dosing regimen. While, the AUC of primaquine calculated from the first dose to 24 hours after the last dose of the weekly dosing regimen was slightly lower compared to the standard dosing regimen. All of the simulated exposures remained well below the safety threshold (red solid line) in G6PD deficiency (please see below figures). We have also included these findings in the discussion section (Line: 424-428).

Line 391: Among the limitations, it would be helpful to comment on the low number of participants in the study, and whether the authors think adding additional subjects would be necessary to support a change in WHO recommendations.

Thank you for this comment. The total number of 21 mother-infant pairs used in this analysis was relatively low. However, the pharmacokinetic sampling schedule was well-design and the model developed for the mothers described the data well in all sample matrices. The uncertainty assessment of all estimated pharmacokinetic parameters showed that the model was robust (RSE < 20%). The parameter estimates were also similar to the values reported in several pharmacokinetic studies in both healthy volunteers and malaria patients. The mother-to-infant model developed in this study also predicted all primaquine concentrations in infants to be below the lower limit of quantification, mirroring what we observed in the actual measurements. We believe that the pharmacokinetic properties of primaquine in breastfeeding women were well described in both plasma and breast milk. We have addressed this in the discussion (Line: 431-435).

Reviewer #2 (Remarks to the Author):

The authors describe the development and application of a population PK model for primaquine (PQ) in lactating mothers and their infants. The manuscript addresses an important topic regarding the use of PQ in breastfeeding mothers. Although the manuscript is well written, several key points should be addressed including the limitations of the model for predictions in infants. Specific comments include:

1. In the methods, much of the text relates to the actual clinical study conducted by Gilder et al. (2018). Whilst I understand that some of the information is required for the population PK model development, it does seem excessive especially as the reference is cited in multiple places.

Thank you for highlighting this. We have revised the manuscript by removing the redundant citation to this reference.

2. The main objective of the study was to develop "a mechanistic pharmacokinetic model, which can be used to predict the exposures of infants to primaquine". However, the model was developed using clinical data from lactating women and applied in infants where there are no clinical/verification data to support the use of the model in this vulnerable population. What confidence do we have that the population PK model can be applied to infants?

Primaquine concentration in infants were measured to be below the lower limit of quantification (1.82 ng/mL), which is reassuring but also the main limitation of this current analysis as we do not have enough data to develop the pharmacokinetic model and estimate pharmacokinetic parameters in infants.

To the best of our knowledge, there is no established pharmacokinetic model in infants that we can adopt to link with the pharmacokinetic model in the mother. However, in the model we used to describe the pharmacokinetics in infants, we accounted for body size (allometric scaling based on body weight), enzyme maturation, and rapid absorption observed in children. These factors primarily contribute to the differences in pharmacokinetic parameters between infants and adults. The developed model produces predictions that align with the observed concentrations, which were all below the lower limit of quantification. The results from our model were also comparable to those presented in the recently published PBPK model (3). In the PBPK model, the median predicted maximum primaquine concentration in infants aged > 28 days receiving oral daily doses in breast milk of 2.98 µg/kg was 0.14 ng/mL, however, this simulation in infants did not account for MAO-A enzyme maturation. In the current analysis, the simulations, assuming 55% enzyme activity at full-term birth, predicted the median maximum primaquine concentration in infants to be 0.31 ng/mL. The median predicted maximum concentration of primaquine was 0.24 ng/mL when excluding the maturation effect from the model. This difference may be partly explained by the method used, wherein the PBPK model assumes more complex tissue compartments. Additionally, the variation in the milk-to-plasma ratio used in the simulation (0.34 in the PBPK model vs. 0.38 in the current analysis) could contribute to this discrepancy.

While direct comparison of model predictions to actual concentrations is not feasible when only LLOQ data are available, the developed mother-to-infant model predicted median primaquine concentrations to be below the LLOQ. This demonstrate that the model's predictions of primaquine concentrations in breastfed infants is close to the actual true values with an absolute maximum error of 1.82 ng/mL (LLOQ), which is expected to be negligible when compared to concentrations observed in adults.

Additional data in neonates and infants administered with primaquine are required to enhance the characterization of pharmacokinetic properties in this population. We also have elaborated this into the discussion section (Line: 436-458)

3. There is considerable uncertainty associated with factors affecting the PQ exposure in infants, including enzyme ontogenies and plasma protein binding (AAG). Whilst the former was accounted for, sensitivity analyses assuming various ontogenies for MAO were not performed. Furthermore, AAG can vary significantly in infants especially in a disease population and affect the exposure of PQ. However, this was not assessed.

The sensitivity analysis on MAO-A enzyme maturation was performed as recommended. The maturation effect implemented in the manuscript assumed that the enzyme activity at full-term birth was 55% compared to adults, based on values reported in the literature, with the enzyme activity reaching 90% at age 4.6 years. In the sensitivity analysis, the enzyme activity at birth was reduced to unrealistic values of 25% and 12.5%, resulting in 90% enzyme activity at 18.9 and 45.0 years of age, respectively. The predicted primaquine concentrations in infants still remained below the lower limit of quantification (1.82 ng/mL). We have mentioned this sensitivity analysis in the manuscript (Line: 229-231, 334-338) and added the plot shown below to the supplementary materials (Supplementary Figure 6).

Regarding the impact of AAG on primaquine exposure, it was not explored in the current analysis due to the lack of available data and the fact that breastfed infants enrolled in the current analysis were healthy (no malaria). The AAG levels in neonates are generally lower compared to adults, which gradually increasing to reach adult levels during adolescence (4). The impact of the change in AAG levels in infants aged 1 month to 2 years old (the age range of infants enrolled in the current study) is expected to be limited. However, we have addressed this aspect in the discussion and referred to the sensitivity analysis conducted in the published PBPK model. (Line: 397-404).

Supplementary Figure 6. Simulated median primaquine concentration in infants with varying MAO-A enzyme activity at full-term birth. The horizontal dashed red line represents the lower limit of quantification of primaquine reported in infants (1.82 ng/mL)

4. The authors did consider that the IDD could change depending on the PQ dose given to mothers or the amount of milk consumed by the infant. However, the impact of pH or fat content which could affect the milk to plasma ratio (M/P) and ultimately the IDD, was not accounted for or discussed.

Thank you for the comment. We acknowledge that breast milk could have a different composition of fat and a varying pH value during different breastfeeding periods. This could impact the distribution of primaquine into the breast milk and thus affect the estimation of milk to plasma ratio. The breast milk data collected in the current study is the mature milk, which has smaller amounts of fat and protein compared to colostrum, and also lower pH value. An additional clinical study is currently under investigation to examine primaquine concentration in colostrum, aiming to better understand drug distribution in neonates (ClinicalTrials.gov Identifier: NCT04984759). We have also addressed and discussed this in the manuscript as recommended (Line: 368-378).

5. The authors did not refer to other mechanistic methods such as PBPK modelling which could be used to predict the exposures of PQ in breastfeeding infants. Indeed, there was a recent publication, where PBPK was used for this very purpose (<https://doi.org/10.1002/psp4.13090>).

Thank you for pointing to this recent publication. We have discussed PBPK modelling and cited this publication in the revised manuscript as recommended.

6. In the conclusion, it is stated that PQ treatment should be considered for mothers with infants aged at least 28 days. If the relevant ontogenies have been applied, why can't the findings be extrapolated to infants < 28 days.

The MAO-A enzyme maturation incorporated in the current analysis primarily describes the changes in enzyme activity in infants compared to adults. The data used in this study were collected from mature milk given to infants aged from 28 days to 2 years. In infants aged < 28 days, breast milk from breastfeeding women exhibits variations in fat and protein components, as well as varying pH during this early breastfeeding period. The effect of open junctions between alveolar cells in the early postpartum period is also hard to predict mathematically. So, we refrain from extrapolating the predictions to infants aged < 28 days as we do not have the data. We hope that data from an additional clinical study, which examines primaquine concentration in colostrum, will become available in the near future. Upon obtaining this data, we can then characterize the distribution of primaquine into breast milk in neonates and estimate the infant daily dose for this specific population. We also have discussed this in the manuscript (Line: 368-378).

References

1. Recht J, Ashley E, White N. Safety of 8-aminoquinolineantimalarial medicines: World Health Organization; 2014. Available from: <https://www.who.int/publications/i/item/9789241506977>.
2. Alving AS, Johnson CF, Tarlov AR, Brewer GJ, Kellermeyer RW, Carson PE. Mitigation of the haemolytic effect of primaquine and enhancement of its action against exoerythrocytic forms of the Chesson strain of *Plasmodium vivax* by intermittent regimens of drug administration: a preliminary report. Bull World Health Organ. 1960;22(6):621-31.
3. Pan X, Abduljalil K, Almond LM, Pansari A, Yeo KR. Supplementing clinical lactation studies with PBPK modeling to inform drug therapy in lactating mothers: Prediction of primaquine exposure as a case example. CPT: pharmacometrics & systems pharmacology. 2023.
4. McNamara PJ, Alcorn J. Protein binding predictions in infants. AAPS PharmSci. 2002;4(1):E4.

Reviewers' Comments:

Reviewer #1:

Remarks to the Author:

Thank you for the modifications, no further comments at this time.

Reviewer #2:

Remarks to the Author:

The authors have addressed my comments thoroughly and appropriately. I have no additional comments.